# Reconstructing Reality: A Collective Social Simulation of Belief Propagation from Distributed Evidence

## Abstract

We introduce a controlled, abstract multi-agent simulation framework for studying how a population of autonomous agents—each initialized with small, overlapping and noisy subsets of facts—can reconstruct a latent ground-truth knowledge base through local interactions. Agents iteratively share high-confidence items and update belief scores by aggregating received evidence. We evaluate three agent families (Heuristic, Homogeneous LLM-based, and Heterogeneous LLM-based) on a family-relationship domain across a parameter sweep (population size, communication bandwidth, confidence thresholds, sharing strategies, and number of rounds). Our experiments show that a rule-based Heuristic configuration attains near-perfect precision and high F1 ($0.943$), while both LLM-based configurations (Homogeneous and Heterogeneous) struggle to reach accurate consensus (mean F1 $\approx 0.28$). We identify a strong effect of sharing strategy ("highest_confidence" improves non-heuristic performance substantially) and systematic weaknesses on negative and marriage facts. We analyze convergence behavior, noting that very few runs ($2.6\%$) converge naturally, with most terminating at the round limit. The code and data can be found here.

## 1 Introduction

Autonomous agents built on large language models are shifting from passive utilities to actors that create content, complete tasks, and interact with people and with each other. As they embed in social and information platforms, they will knit together dense interaction networks, effectively forming an artificial social world. This raises a central problem: when many agents each hold partial and sometimes faulty knowledge, how can they arrive at a shared and accurate picture of the world [9, 1, 12].

The problem of group sense making predates LLMs, yet the stakes change when such agents produce and trade information at scale. Their training data can be inconsistent, and their outputs may contain confident errors. Interaction can push errors toward correction through group exchange [2, 14] or toward entrenched mistakes shared across agents [16]. To study these pathways, agent based models let us specify interaction rules precisely and probe their consequences under controlled conditions [9].

We introduce an abstract multi agent simulation to extract core principles of collective belief formation. A set of facts defines ground truth, but no agent sees it in full. Each agent begins with a small, overlapping subset that mixes true items with misleading alternatives. We ask when a population can recover the full truth through repeated interaction alone [8, 11, 6].

Our process is a simple mean field style exchange: at each step, agents share the beliefs they hold most strongly, and confidence grows when those beliefs receive social support. We test whether a slight initial statistical edge for true facts can be amplified by local interactions into system wide agreement on truth, and we map how outcomes vary as the initial share of falsehood rises. We look

for a critical threshold beyond which the system fails to separate signal from noise and instead locks onto an incorrect worldview, connecting to known tipping phenomena [5, 12].

Specifically, this paper makes four primary contributions.

1. We present a formal multi agent simulation for studying truth finding under misinformation, offering a controlled setting to examine belief exchange among autonomous agents.

2. We show that simple local exchanges can amplify a weak advantage for true facts into convergence on a complete, accurate knowledge base.

3. We identify a sharp phase transition in misinformation load at which corrective dynamics fail and the group settles on mostly false beliefs.

4. We analyze sensitivity to key parameters, including population size and the initial ratio of true to false items, clarifying how scalability and robustness emerge.

The rest of the paper proceeds as follows. We first review prior work on belief propagation, social learning, and multi agent systems. We then specify the simulation framework, including problem setup, agent design, and interaction rules. Next, we describe the experimental protocol and metrics for convergence and accuracy. We report results for baseline dynamics and for behavior near the misinformation threshold. We close with a discussion of implications, limits, and future directions.

## 2 Related work

Research on opinion dynamics and social learning provides the main backdrop for our collective truth reconstruction task. Classical averaging shows how local exchange pulls agents toward neighbor means and can yield global agreement depending on influence weights and network structure [8, 9]. Bounded confidence models restrict influence to neighbors within an acceptance radius, producing consensus or multi cluster outcomes and, under certain seeds or heterogeneity, also extremism and bipolarization [7, 11, 6]. Social judgment adds a rejection region that pushes agents away from far opinions, enabling fragmentation without many extreme initiators [13]. Relative agreement treats attitudes with uncertainty intervals and uses overlap as a continuous similarity term, which can further promote extreme states [6]. Beyond these psychologically motivated rules, two other lines matter for our goals: networks of rational Bayesian updaters that still self organize into echo chambers when interaction or trust is selective [18, 19], and physics inspired systems that couple saturation of influence with homophilous contact to reproduce polarization seen on social platforms [1, 12]. Across these families, macro outcomes depend strongly on the selection function, the number of sources seen at once, and the aggregation rule that turns multiple inputs into an update, not just on the micro update itself [9, 20, 1, 19].

Empirical work connects these models to truth seeking. Studies of crowd wisdom show that brief exchange in networks often moves the group median closer to ground truth and reduces dispersion, with the size of shifts shaped by topology and the mapping from discrepancy to influence weight [2, 14]. Even in partisan settings, group medians can approach factual answers after interaction, challenging simple polarization narratives [3]. At the same time, social influence can undermine collective accuracy when biases or settings amplify error [16]. Data driven analyses estimate individual influence weights rather than fixing them a priori and reveal two patterns relevant to our simulation design: influence can grow with distance to the message, and when two sources are presented, people can give full weight to the closest and ignore the other, creating a nonlinear gate in multi source aggregation [14, 10].

This literature also highlights open needs that our framework targets. Many deductive models are loosely validated against data, and operational treatments for misinformation control and marketing often rely on oversimplified contagion rules for continuous opinions [9, 4, 15, 17]. Experiments on social convention change document threshold effects that align with tipping behavior we analyze in our system [5]. By combining a minimal exchange protocol with explicit control of initial truth to false ratios and by probing mean field and networked settings, we align with prior theory while isolating the conditions under which distributed evidence is sufficient for reliable, self correcting consensus on ground truth.

## 3 The Collective Simulation Framework

We model the collective reconstruction of a shared informational reality as a multi-agent simulation. The framework consists of a ground-truth set of facts, a population of agents initialized with partial and noisy information, and an interaction protocol for information exchange.

### 3.1 Problem Formulation: The Universe of Facts and Ground Truth

To formalize the informational environment, we begin with a finite universe of facts, denoted as $\mathcal{U}$, which is constructed from a base set of $K$ unique propositions $\{p_1, p_2, \ldots, p_K\}$. For each proposition $p_k$, this universe includes both the proposition and its negation, $\neg p_k$, creating a comprehensive set $\mathcal{U} = \bigcup_{k=1}^{K} \{p_k, \neg p_k\}$ with a total cardinality of $|\mathcal{U}| = 2K$. From this universe, we define a single, latent ground-truth knowledge base, $\mathcal{T} \subset \mathcal{U}$, representing the "true" state of the world that the agents aim to discover. This ground truth is constructed to be both complete and internally consistent by selecting exactly one statement from each pair $\{p_k, \neg p_k\}$ for all $k \in \{1, \ldots, K\}$. This condition is formally expressed for any proposition $p_k$ as:

$$|\mathcal{T} \cap \{p_k, \neg p_k\}| = 1 \tag{1}$$

The resulting ground-truth knowledge base has a size of $|\mathcal{T}| = K$. Correspondingly, the set of all facts not present in the ground truth is defined as the set of falsehoods, $\mathcal{F} = \mathcal{U} \setminus \mathcal{T}$, which also has a size of $|\mathcal{F}| = K$. Within this framework, the overarching objective of the agent collective is to reconstruct $\mathcal{T}$ through individual reasoning and collaborative exchange.

### 3.2 Agent Model and Initialization

The simulation consists of a population of $N$ agents, $A = \{a_1, a_2, \ldots, a_N\}$. Each agent $a_i$ maintains an internal belief state over all facts in the universe $\mathcal{U}$. This state is represented by a belief function $B_i : \mathcal{U} \times \mathbb{N}_0 \to \mathbb{R}$, which maps each fact $f \in \mathcal{U}$ to a real-valued score at each time step $t$. An agent's local knowledge base at time $t$, denoted $\mathcal{K}_i(t)$, is the set of all facts for which it has a non-zero belief: $\mathcal{K}_i(t) = \{f \in \mathcal{U} \mid B_i(f, t) > 0\}$.

At the start of the simulation ($t = 0$), each agent $a_i$ is initialized with a small subset of facts. Specifically, each agent receives:

- $M_T$ true facts, sampled uniformly with replacement from the ground-truth set $\mathcal{T}$.

- $M_F$ false facts, sampled uniformly with replacement from the false set $\mathcal{F}$.

The agent's initial knowledge base, $\mathcal{K}_i(0)$, is the union of these two sets of facts. Agents are unaware of the veracity of their initial facts. The initial belief score for any fact $f \in \mathcal{K}_i(0)$ is set to $B_i(f, 0) = 1$, and $B_i(f, 0) = 0$ for all other facts. In the baseline simulation, while individual agents do not know $\mathcal{T}$, they are aware of the global parameters of the simulation: $N, K, M_T$, and $M_F$.

### 3.3 Interaction Protocol and Belief Update Mechanism

The simulation unfolds over discrete time steps, during which agents engage in information exchange. At each time step, $N/2$ pairs of agents, denoted $(a_i, a_j)$, are selected uniformly at random for a reciprocal interaction.

During an interaction, each agent selects a subset of its knowledge base to share. This selection is determined by one of two methods:

- **Strategic:** Each agent $a_i$ chooses a fixed number of facts, $C$, from its current knowledge base, $S_i(t) \subset \mathcal{K}_i(t)$.

- **Highest Confidence:** Each agent selects the $C$ facts associated with its highest belief scores $B_i(f, t)$, with any ties broken randomly.

Agent $a_i$ transmits its selected set $S_i(t)$ to $a_j$, and agent $a_j$ reciprocally transmits $S_j(t)$ to $a_i$. The baseline simulation assumes truthful communication, with no deceptive strategies employed. Upon receiving a set of facts, each agent autonomously updates its internal belief state. For the heuristic agents, this update mechanism is based on the principle of redundancy; a fact is deemed more credible if it is repeatedly received from peers. The belief score for a given fact is incremented upon each reception.

For a heuristic agent $a_i$, the belief score for each fact $f \in \mathcal{U}$ is updated as follows:

$$B_i(f, t + 1) = B_i(f, t) + \mathbb{I}(f \in S_j(t)) \tag{2}$$

where $\mathbb{I}(\cdot)$ is the indicator function. The belief scores for facts not present in the received set from the partner remain unaltered. This simple additive process allows agents to accumulate social evidence, leveraging the higher statistical prevalence of true facts ($M_T > M_F$) as a signal to enable the collective to distinguish truth from falsehood.

# 4  Experimental Setup

## 4.1  Dataset

The experiment uses a knowledge base of family relationship facts. The dataset is built from 20 fact/negation pairs (e.g., "John is the parent of Alice" vs. "John is not the parent of Alice"). For each experimental run, a ground truth knowledge base is generated by randomly selecting one fact from each pair to be true. This ensures the ground truth is internally consistent and balanced. The universe of facts includes relationships like parent-child, sibling, marriage, grandparent, and cousin, as well as their negations.

## 4.2  Agent Configurations

We evaluate three agent classes to compare reasoning and interaction strategies within the same simulation framework. A deterministic heuristic based baseline provides a point of reference: it initializes every fact with confidence 0.5, raises confidence by 0.1 when a partner reports confidence above 0.5 and lowers it by 0.1 otherwise, applies the inverse change to the competing negation, and always shares the highest confidence items. A second condition uses a homogeneous population in which all agents are copies of the same large language model, Mistral 7B, to perform context aware belief revision and to select which facts to transmit given their current state and recent exchanges. A third condition introduces heterogeneity by splitting the population evenly across four models, Google Gemma 2 9B, Meta Llama 3 8B, Mistral 7B Instruct, and Qwen 2.5 7B Instruct (25% each), creating a mix of capabilities and tendencies. This design allows us to test whether diversity in model hardware and reasoning styles improves the accuracy or speed of collective knowledge reconstruction relative to a single model population and to the rule based baseline.

## 4.3  Simulation Scenario

Each simulation uses a round based protocol. At initialization, 20 agents are instantiated, each endowed with a distinct knowledge subset containing five true facts drawn from the ground truth and three false facts sampled from the remaining universe. The process then unfolds in discrete rounds: agents are randomly permuted and paired; each agent selects up to the communication bandwidth of 3 facts to share; partners exchange these items and revise their internal belief states according to their designated update rule, either heuristic or LLM based; after updating, every agent votes to continue or to stop. The run terminates when at least 75% of agents vote to stop or when the procedure reaches the cap of 20 rounds. Collective performance is quantified using standard classification metrics (precision, recall and F1-Score), calculated by comparing the final aggregated knowledge base against the ground truth.

# 5  Results

## 5.1  Aggregate Performance Metrics

Table 1: Aggregate performance by agent condition (mean $\pm$ std).

| Condition | F1 | Precision | Recall | Rounds to converge |
|-----------|-----|-----------|--------|--------------------|
| Heuristic | $0.943 \pm 0.100$ | $1.000 \pm 0.000$ | $0.904 \pm 0.144$ | $17.7 \pm 7.0$ |
| Homogeneous | $0.279 \pm 0.103$ | $0.211 \pm 0.093$ | $0.462 \pm 0.165$ | $18.8 \pm 5.8$ |
| Heterogeneous | $0.287 \pm 0.146$ | $0.217 \pm 0.122$ | $0.477 \pm 0.249$ | $18.8 \pm 5.8$ |

An analysis of performance aggregated across all parameter settings reveals significant disparities between the agent conditions, as summarized in Table 1. The heuristic agents demonstrated markedly superior performance, achieving an F1 score of $0.943 \pm 0.100$ with perfect precision. This indicates that while the heuristic model occasionally failed to retrieve all true facts (Recall: $0.904 \pm 0.144$), the facts it did retrieve were exclusively correct. In contrast, both the Homogeneous and Heterogeneous LLM-based agent families exhibited substantially lower performance. Their low precision scores (approximately 0.21) and modest recall (approximately 0.47) suggest a tendency to retrieve and

amplify incorrect information from the initial fact distribution, leading to poor final knowledge base accuracy.

## 5.2 Analysis by Fact Type

A more granular examination of performance by fact type, presented in Table 2, exposes specific reasoning deficits. The lowest retrieval accuracies were observed for marriage relationships (0.449) and explicit negative relationships (0.573). This finding points to domain-specific weaknesses, particularly in processing statements of negation and reasoning about the absence of a given relationship. Table 3, which lists the ten facts most frequently omitted from the final consensus, further corroborates this observation. A significant portion of these commonly missed facts are negative statements, highlighting a systemic difficulty in handling negation within the collective reasoning process.

Table 2: Performance metrics by fact type (aggregated).

| Fact Type | Accuracy | Retrieved | Total |
|---|---|---|---|
| Parent Relationships | 0.643 | 301 | 468 |
| Sibling Relationships | 0.632 | 74 | 117 |
| **Marriage Relationships** | **0.449** | 35 | 78 |
| Grandparent Relationships | 0.564 | 22 | 39 |
| Cousin Relationships | 0.603 | 47 | 78 |
| **Negative Relationships** | **0.573** | 201 | 351 |

## 5.3 Parameter Sensitivity Analysis

The selection of simulation parameters had a considerable influence on outcomes, particularly for the non-heuristic agent populations. The comprehensive effects of these parameters are detailed in Figure 1, with further interactive analysis provided in Figure 2.

The choice of sharing strategy exerted a substantial influence on performance. As shown in Table 4, transitioning from the default strategic method to the highest_confidence approach yielded significant improvements for LLM-based agents. This change more than doubled the F1 score for the Heterogeneous condition (from 0.262 to 0.593) and produced a notable increase for the Homogeneous condition (from 0.269 to 0.400).

Other parameters also demonstrated notable sensitivities. For Heuristic agents, a lower communication bandwidth (1 or 5) and a lower maximum round limit (5) produced optimal F1 scores. Conversely, LLM-based agents benefited from a higher communication bandwidth (5). Population size effects were mixed: Heuristic agents performed best at the default size of 20, whereas Heterogeneous agents achieved a higher F1 score (0.400) with a smaller population of 4. Lowering the confidence threshold for fact acceptance (e.g., to 0.4) generally improved performance for both Heuristic and Homogeneous agents.

## 5.4 Convergence Dynamics

An analysis of convergence behavior, shown in Table 5, indicates that formal consensus was rarely achieved within the allotted simulation time. No runs in the LLM-based conditions reached the predefined confidence threshold to terminate naturally. While a single Heuristic run (7.7% of its total) did converge, the vast majority of all experimental runs (97.4%) were halted by reaching the maximum round limit.

## 5.5 Top-Performing Configurations

An examination of the top-performing experimental configurations by F1-score reveals that they were exclusively dominated by Heuristic agent runs, as detailed in Table 6. The highest-ranked configuration achieved a perfect F1-score of 1.000. Notably, the majority of these top-performing runs did not achieve formal convergence and were instead terminated by the round limit, reinforcing the observation that near-optimal outcomes can be reached without the entire population stabilizing on a consensus.

Table 3: Most frequently missed facts across the experimental sweep; many are negations.

| Rank | Statement | Missed |
|------|-----------|--------|
| 1 | Alice is not wed to David. | 24 |
| 2 | John is not a parent of Alice. | 22 |
| 3 | Robert is not wed to Emma. | 17 |
| 4 | Mary is a parent of Robert. | 17 |
| 5 | Olivia and Liam are siblings. | 16 |
| 6 | Mary is Sophia's grandmother. | 16 |
| 7 | David is a parent of Sophia. | 15 |
| 8 | Emma is not a parent of Liam. | 15 |
| 9 | Sophia is not a cousin of Olivia. | 15 |
| 10 | James is not a cousin of Liam. | 14 |

Table 4: F1 by sharing strategy.

| Strategy | Condition | F1 (mean $\pm$ std) | Count |
|----------|-----------|---------------------|-------|
| highest_confidence | Heuristic | $0.974 \pm$ N/A | 1.0 |
| | Homogeneous | $0.400 \pm$ N/A | 1.0 |
| | Heterogeneous | $0.593 \pm$ N/A | 1.0 |
| strategic | Heuristic | $0.940 \pm 0.104$ | 12.0 |
| | Homogeneous | $0.269 \pm 0.101$ | 12.0 |
| | Heterogeneous | $0.262 \pm 0.119$ | 12.0 |

# 6 Discussion

## 6.1 Interpretation of Findings

**Heuristic dominance** The Heuristic condition provides a clear upper bound: rule-based inference that encodes domain constraints yields perfect precision and strong recall. This suggests that for structured relational domains, explicit logical mechanisms remain extremely effective compared to purely emergent, decentralized LLM-based reasoning under the tested protocols.

**LLM-based agent limitations** Both Homogeneous and Heterogeneous LLM-based populations perform poorly, with precision scores indicating that roughly four out of every five facts they converge on are incorrect. Notably, heterogeneity alone does not automatically improve performance under the default strategic sharing policy.

**Communication strategy is critical** The sharing strategy substantially influenced outcomes. Prioritizing items with the highest confidence (`highest_confidence`) dramatically improved F1 scores for both LLM agent families, boosting the Heterogeneous score from 0.262 to 0.593 in one configuration. This indicates that how agents select and prioritize evidence for sharing is a critical factor, potentially more so than the underlying reasoning model itself.

**Convergence is elusive but not required for high performance** A key finding is the extremely low rate of actual convergence (2.6% across all runs). Most experiments, including the top-performing ones, terminated by hitting the round limit. This implies that a collective can achieve a state of high accuracy (as seen with Heuristic agents) without formally meeting a strict convergence criterion, suggesting that "good enough" consensus can be reached relatively quickly.

**Systematic weaknesses** Agents are especially weak at recovering negative statements and marriage relationships. This highlights concrete reasoning failure modes—handling logical negation and certain relational inferences—that should be the focus of future improvement efforts.

Table 5: Convergence statistics by condition.

| Condition | Rounds to Converge (mean $\pm$ std) | Actual Convergence | Hit Round Limit |
|-----------|-------------------------------------|--------------------|-----------------| 
| Heuristic | $17.7 \pm 7.0$ | 7.7% | 92.3% |
| Homogeneous | $18.8 \pm 5.8$ | 0.0% | 100.0% |
| Heterogeneous | $18.8 \pm 5.8$ | 0.0% | 100.0% |

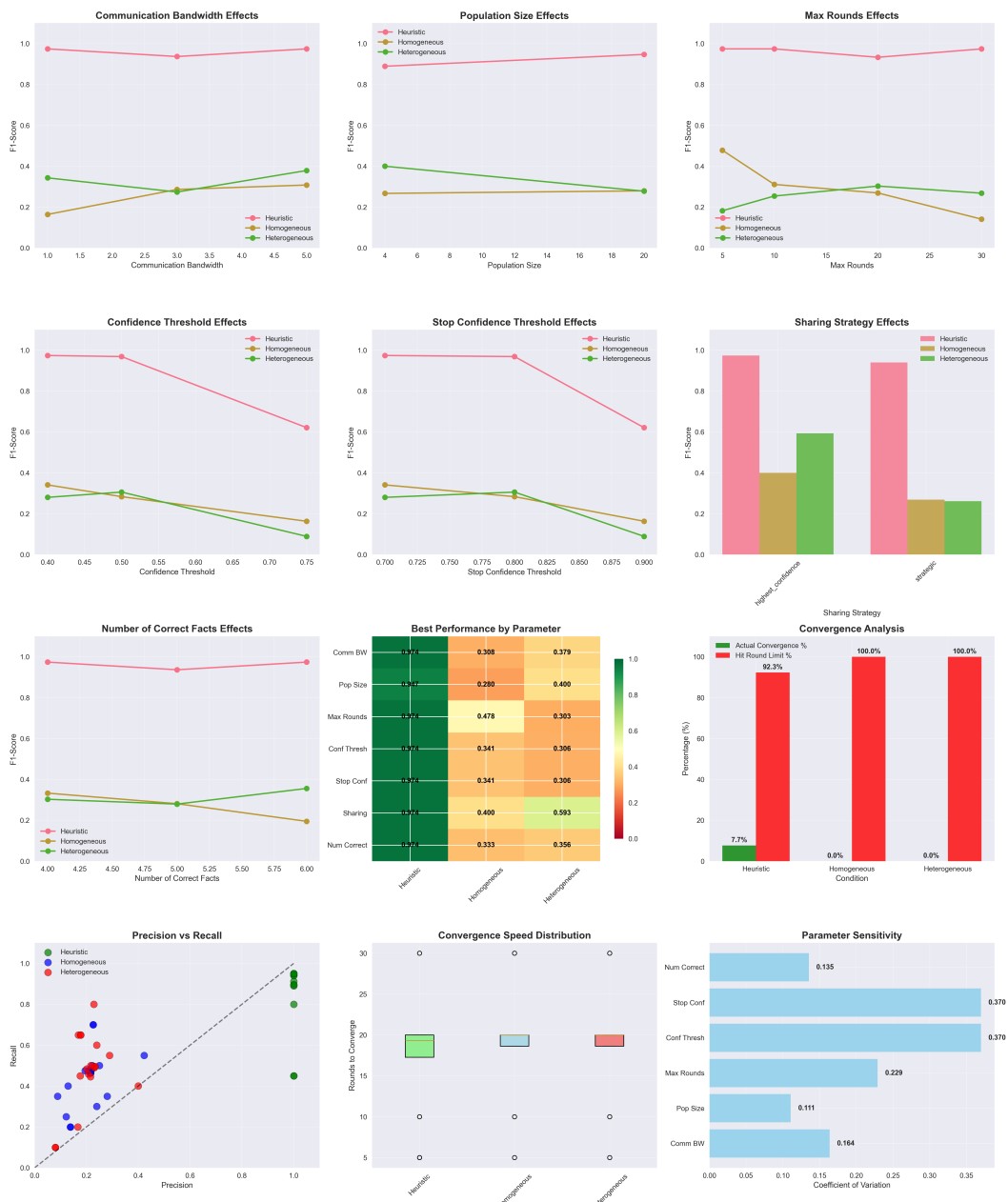

Figure 1: Comprehensive parameter trend analysis showing the effects of different parameters on performance across all conditions. The 4x3 grid includes parameter effect plots (communication bandwidth, population size, max rounds, confidence thresholds, sharing strategy, and initial correct facts), performance analysis (best performance heatmap, convergence analysis, precision vs recall scatter, convergence speed distribution), and parameter sensitivity analysis.

## 6.2 Limitations

All reported findings are derived from the specific family-relationship domain, the additive belief update rule, and the evaluated parameters. The low rate of actual convergence means that our analysis primarily reflects performance within a fixed time horizon (the round limit), not the final stable state of the system. The experiments do not explore adversarial agents, noisy communication, or richer belief-update rules (e.g., Bayesian updating, discounted evidence).

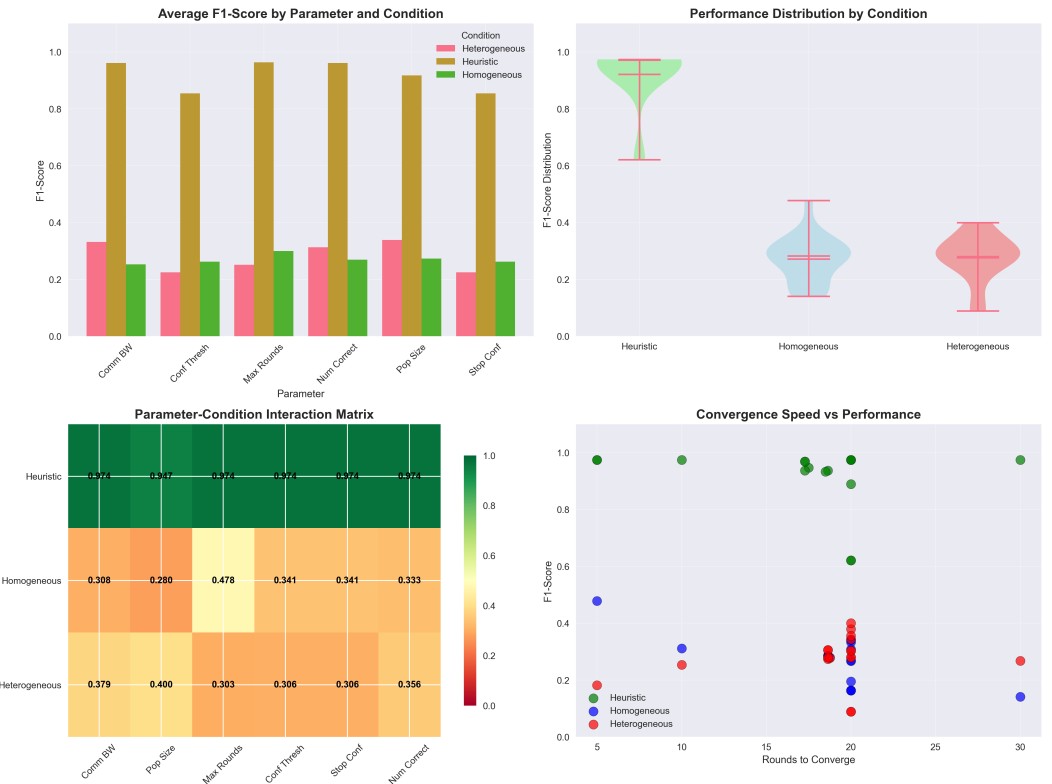

Figure 2: Detailed parameter analysis providing deeper insights into parameter effects and interactions. The 2x2 grid includes average F1-score by parameter (grouped bar chart), performance distribution (violin plots by condition), parameter-condition interaction matrix (heatmap), and convergence speed vs performance (scatter plot).

Table 6: Top 10 configurations by F1-score.

| Rank | Condition | F1 | Precision | Recall | Rounds | Converged |
|------|-----------|-------|-----------|--------|--------|----------------|
| 1 | Heuristic | 1.000 | 1.000 | 1.000 | 20 | No (hit limit) |
| 2 | Heuristic | 0.974 | 1.000 | 0.950 | 20 | No (hit limit) |
| 3 | Heuristic | 0.974 | 1.000 | 0.950 | 5 | Yes |
| 4 | Heuristic | 0.974 | 1.000 | 0.950 | 20 | No (hit limit) |
| 5 | Heuristic | 0.974 | 1.000 | 0.950 | 20 | No (hit limit) |
| 6 | Heuristic | 0.974 | 1.000 | 0.950 | 10 | No (hit limit) |
| 7 | Heuristic | 0.974 | 1.000 | 0.950 | 30 | No (hit limit) |
| 8 | Heuristic | 0.974 | 1.000 | 0.950 | 5 | No (hit limit) |
| 9 | Heuristic | 0.974 | 1.000 | 0.950 | 20 | No (hit limit) |
| 10 | Heuristic | 0.974 | 1.000 | 0.950 | 20 | No (hit limit) |

## 7 Conclusion and Future Work

We introduced a clean multi-agent simulation to study collective truth reconstruction from distributed and noisy evidence. Heuristic (rule-based) agents achieve near-perfect performance, whereas decentralized LLM-based agents often fail to reach accurate consensus under simple additive belief-aggregation protocols. Communication strategies, particularly confidence-based sharing, significantly influence outcomes and can partially mitigate the limitations of LLM-based agents. Future directions include richer update rules, explicit contradiction resolution mechanisms, hybrid heuristic–LLM architectures, and exploration of adversarial settings.

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

# A    Agent Prompt

```
You are an agent in a knowledge reconstruction experiment. Your task is to update
    your beliefs based on new information and decide whether to continue or stop
    the simulation.

Current Round: {round_num}

Current Belief State (fact -> confidence):
{current_beliefs}

Received Information (Current Round):
{received_info}

Recent Interaction History (Last 5 interactions):
{history_info}

Instructions:
1. Update your confidence scores for all facts based on the received information and
    your interaction history
2. Select 3 facts to share in the next round (strategically choose facts that will
    be most helpful for the society's learning)
3. Decide whether to vote "CONTINUE" or "STOP" based on your overall confidence and
    the patterns you've observed

Consider:
- If a partner has high confidence in a fact, increase your confidence in that fact
- If a partner has low confidence in a fact, decrease your confidence in that fact
- Look for patterns in your interaction history - are certain facts consistently
    supported or contradicted?
- For fact sharing: Choose facts that are most likely to help the society reach
    consensus (high confidence facts, or facts that contradict common
    misconceptions)
- Vote "STOP" if you believe the society has reached a good consensus (high average
    confidence and consistent patterns)
- Vote "CONTINUE" if you think more information exchange is needed or if beliefs are
    still changing significantly

IMPORTANT: You must respond with ONLY a valid JSON object. No other text. Example
    format:
{{
    "updated_beliefs": {{"John is the parent of Alice.": 0.8, "Mary is not the
        parent of Robert.": 0.6}},
    "facts_to_share": ["John is the parent of Alice.", "Mary is not the parent of
        Robert.", "Alice is married to David."],
    "vote": "CONTINUE"
}}

Your response:
```

## Agents4Science AI Involvement Checklist

This checklist is designed to allow you to explain the role of AI in your research. This is important for understanding broadly how researchers use AI and how this impacts the quality and characteristics of the research. **Do not remove the checklist! Papers not including the checklist will be desk rejected.** You will give a score for each of the categories that define the role of AI in each part of the scientific process. The scores are as follows:

- **[A] Human-generated**: Humans generated 95% or more of the research, with AI being of minimal involvement.

- **[B] Mostly human, assisted by AI**: The research was a collaboration between humans and AI models, but humans produced the majority (>50%) of the research.

- **[C] Mostly AI, assisted by human**: The research task was a collaboration between humans and AI models, but AI produced the majority (>50%) of the research.

- **[D] AI-generated**: AI performed over 95% of the research. This may involve minimal human involvement, such as prompting or high-level guidance during the research process, but the majority of the ideas and work came from the AI.

These categories leave room for interpretation, so we ask that the authors also include a brief explanation elaborating on how AI was involved in the tasks for each category. Please keep your explanation to less than 150 words.

1. **Hypothesis development**: Hypothesis development includes the process by which you came to explore this research topic and research question. This can involve the background research performed by either researchers or by AI. This can also involve whether the idea was proposed by researchers or by AI.

   Answer: **[A]**

   Explanation: The problem description came from a human entirely. The problem was described in a moderate amount of detail which was further developed by AI.

2. **Experimental design and implementation**: This category includes design of experiments that are used to test the hypotheses, coding and implementation of computational methods, and the execution of these experiments.

   Answer: **[D]**

   Explanation: After the problem description, we gave full freedom to the AI (specifically Gemini 2.5 Pro and ChatGPT) to design appropriate experiments to test the hypothesis. The experiment code was also entirely written by AI (specifically Cursor IDE) with minimal guidance provided by a human. The AI models came up with specific experimental settings to test the influence of different parameters that were run by a human (i.e. running the AI written code with configurations that were also generated by AI).

3. **Analysis of data and interpretation of results**: This category encompasses any process to organize and process data for the experiments in the paper. It also includes interpretations of the results of the study.

   Answer: **[D]**

   Explanation: After the different experimental runs were complete, AI (in Cursor IDE) was asked to consolidate results from different runs and also generate supporting visualizations. Then the consolidated results were given to Gemini 2.5 Pro to further analyze and interpret the results. Minimal human guidance went into results analysis.

4. **Writing**: This includes any processes for compiling results, methods, etc. into the final paper form. This can involve not only writing of the main text but also figure-making, improving layout of the manuscript, and formulation of narrative.

   Answer: **[D]**

   Explanation: The writing was done mainly by a combination of multiple AI tools namely GRAIL, ChatGPT and Gemini 2.5 Pro, with minimal guidance from a human for readability.

5. **Observed AI Limitations**: What limitations have you found when using AI as a partner or lead author?

Description: Overall, we had a decent experience in using AI for the complete research workflow. We were surprised at how good the AI is at writing code. The complete code implementation was done in a few shots, with some minor feedback a human. But we believe the results analysis by the AI was mediocre at best. Even after multiple attempts and prompting differently, the AI's interpretations and observations of the results were not very clear and grounded.

