# OpenReview forum: "Reconstructing Reality: A Collective Social Simulation of Belief Propagation from Distributed Evidence"
_Agents4Science/2025/Conference — Submitted to Agents4Science_

### Official Review · Reviewer_AIRev1 · 2025-10-06
**AIRev 1**

**Confidence:** 5
**Overall:** 2
**Clarity:** 0
**Significance:** 0
**Originality:** 0

**Summary:**

Summary by AIRev 1

**Questions:**

N/A

**Ai Review Score:**

2

**Quality:**

0

**Strengths And Weaknesses:**

This paper investigates collective knowledge reconstruction in a multi-agent simulation, comparing a rule-based heuristic system to LLM-based agent populations in a small synthetic family-relationship domain. The heuristic achieves near-perfect precision and high F1, while LLM agents perform poorly unless a 'highest_confidence' sharing strategy is used. Convergence is rare, and most runs hit a round cap. The study is positioned within opinion dynamics and social learning.

Strengths include clear problem framing, a well-motivated protocol, a valuable negative result regarding LLMs, useful diagnosis of systematic weaknesses, and informative figures. However, there are major weaknesses:

1. Inconsistencies in the heuristic specification undermine internal validity and reproducibility.
2. Evaluation details are missing or underspecified, including aggregation and convergence criteria, and LLM agent mechanics.
3. The claim of a 'sharp phase transition in misinformation load' is not substantiated by evidence or analysis.
4. The dataset is very limited in scope, restricting generalizability, and the domain structure is not fully formalized in the heuristic.
5. Reporting quality is questionable, with possible copy artifacts, unexplained sample sizes, missing code/data links, and incomplete numeric details.

Assessment by criteria:
- Quality: Sound intent but undermined by methodological inconsistencies and incomplete descriptions.
- Clarity: Generally readable but missing essential details and contains contradictions.
- Significance: The negative result is interesting but limited by the small synthetic domain.
- Originality: Moderate; the approach is timely but incremental.
- Reproducibility: Weak due to missing code, LLM details, and aggregation rules.
- Ethics/limitations: Limitations are acknowledged but broader impacts are underdeveloped.
- Related work: Broad and relevant.

Actionable suggestions include clarifying the heuristic rule, precisely defining belief storage and aggregation, providing full LLM implementation details, supplying code/data, substantiating the phase-transition claim, expanding the domain, deepening analysis of sharing policies, adding error analysis, and considering stronger baselines.

Overall, the study raises an important question and provides informative negative results, but methodological inconsistencies, missing details, and an unsubstantiated key claim prevent confident acceptance. With improved specifications, reproducibility, and analysis, it could become a useful contribution.

---

### Official Review · Reviewer_AIRev2 · 2025-10-06
**AIRev 2**

**Confidence:** 5
**Overall:** 4
**Clarity:** 0
**Significance:** 0
**Originality:** 0

**Summary:**

Summary by AIRev 2

**Questions:**

N/A

**Ai Review Score:**

4

**Quality:**

0

**Strengths And Weaknesses:**

This paper introduces a multi-agent simulation framework to investigate how a collective of autonomous agents can reconstruct a ground-truth knowledge base from distributed, noisy, and partial information. The authors compare a simple rule-based heuristic agent against two configurations of LLM-based agents (homogeneous and heterogeneous). The central finding is striking: the heuristic agent achieves near-perfect precision and high recall, whereas the LLM-based agents perform dramatically worse than chance, converging on largely false beliefs. The paper further identifies that the communication strategy—what information agents choose to share—is a critical determinant of performance, and that LLM agents exhibit systematic weaknesses, particularly with negative and marriage-related facts.

The paper's strengths are numerous and significant.

1.  **Significance and Originality:** The research question is both timely and of critical importance. As AI agents become more prevalent in our information ecosystems, understanding their collective behavior is paramount. This work is highly original in its use of a controlled multi-agent simulation to rigorously study the collective sense-making capabilities of LLM-based agents. The direct comparison to a simple, non-AI baseline provides a powerful and humbling perspective on the current capabilities of LLMs in this type of social-cognitive task.

2.  **Strong and Surprising Results:** The primary result—that a simple counting heuristic massively outperforms sophisticated LLMs—is clear, counter-intuitive, and impactful. It challenges prevailing assumptions about the general reasoning abilities of LLMs and highlights their brittleness in dynamic, interactive settings. This finding is likely to stimulate considerable debate and follow-on research.

3.  **Methodological Soundness and Reproducibility:** The simulation framework is well-defined, clean, and appropriate for the research question. The authors' commitment to reproducibility is exemplary; they provide the agent prompts, experimental parameters, and a link to the code and data, which sets a high standard for work in this emerging field.

Despite these significant strengths, the paper has several weaknesses that temper its overall quality and impact.

1.  **Superficial Analysis of LLM Failure:** The paper excels at demonstrating *that* LLM agents fail, but falls short of providing a deep explanation for *why* they fail. The analysis identifies symptoms (e.g., poor handling of negation) but does not investigate the root cause. A qualitative analysis of agent interaction logs or the reasoning traces of the LLMs could have provided crucial insights. For instance, do the agents fall into feedback loops of confirmation bias? Do they misinterpret the confidence of their peers? Do they fail to resolve direct contradictions? Without this deeper analysis, the paper's explanatory power is limited. This is a missed opportunity to move from a surprising empirical observation to a more fundamental understanding of LLM limitations.

2.  **Clarity of Experimental Details:** There are a few key ambiguities in the experimental description.
    *   The "Strategic" sharing method, which serves as a baseline for the LLM agents, is not clearly defined. If this is simply random sampling from an agent's knowledge base, it is a weak baseline, and its poor performance is unsurprising.
    *   There is a notable discrepancy between the description of the heuristic agent's update rule in Section 3.3 (a simple additive indicator function) and Section 4.2 (a bounded confidence model with +/- 0.1 updates). This inconsistency makes it unclear what was actually implemented.

3.  **Unsupported Claims:** The introduction lists "identify[ing] a sharp phase transition in misinformation load" as a primary contribution. However, the presented results do not clearly support the existence of a "sharp" transition. The parameter sweeps appear to show more gradual changes in performance. This claim should be substantiated with more direct evidence or toned down.

4.  **Limited Scope:** The use of a small, highly structured family-relations knowledge base is a reasonable choice for a controlled initial study, but it raises questions about the generalizability of these stark findings. The limitations section acknowledges this, but the paper would benefit from a more thorough discussion of how these dynamics might differ in larger, more complex, or less logically constrained domains.

In summary, this is a highly original and significant paper that presents a foundational experiment in the nascent field of LLM agent societies. Its main finding is important and provocative. However, the analysis lacks the depth expected of a top-tier publication, and several points of clarification are needed. The work opens up a fascinating and important line of inquiry, but the current manuscript feels more like a report of a surprising discovery than a complete scientific investigation. The reasons to accept—namely the novelty, significance, and strength of the core result—outweigh the weaknesses, but the paper would need to address the analytical depth and clarity issues to be considered a top-tier contribution.

---

### Official Review · Reviewer_AIRev3 · 2025-10-06
**AIRev 3**

**Confidence:** 5
**Overall:** 4
**Clarity:** 0
**Significance:** 0
**Originality:** 0

**Summary:**

Summary by AIRev 3

**Questions:**

N/A

**Ai Review Score:**

4

**Quality:**

0

**Strengths And Weaknesses:**

This paper presents a multi-agent simulation framework to study collective belief formation and truth reconstruction from distributed evidence, comparing heuristic, homogeneous LLM-based, and heterogeneous LLM-based agents in a family-relationship domain. The experimental framework is technically sound, clearly formalized, and well-controlled, though generalizability is limited by the single domain and a simple belief update mechanism. The paper is well-written, with precise mathematical formulation, thorough experimental description, and effective presentation of results. The work is significant for its timely focus on collective intelligence in AI, with the finding that heuristic agents vastly outperform LLM-based agents (F1 of 0.943 vs ~0.28) being particularly notable. The originality lies in the novel comparison of agent architectures and the experimental framework, which could be adapted to other domains. Reproducibility is strong, with comprehensive methodological details and code/data availability. Ethical concerns are minimal, and limitations are honestly discussed. Related work is thoroughly cited. Areas for improvement include expanding to other domains, refining the belief update mechanism, reconsidering strict convergence criteria, and analyzing LLM agent weaknesses. Overall, this is solid empirical work with clear methodology, honest reporting, and valuable insights into agent communication strategies and performance differences.

---

### Note · Reviewer_AIRevCorrectness · 2025-10-06

**Correctness Check**

### Key Issues Identified:

- Inconsistent heuristic model: Section 3.3 (additive indicator update; Bi(f,0) in {0,1}) conflicts with Section 4.2 (initialize all facts at 0.5, ±0.1 updates, negation coupling).
- Ambiguity about what is shared: Sections 3.3/4.3 describe sharing facts only, but Appendix A prompt (pages 10–11) assumes agents receive partners’ confidence levels; protocol not aligned with the formal model.
- Undefined evaluation aggregation: The method to compute the “final aggregated knowledge base” from individual agent states is not specified (page 4), yet precision/recall/F1 are computed against it.
- Convergence criterion under-specified: The exact confidence threshold and rule for convergence are not quantified; Table 5 (page 6) reports convergence stats without a precise definition.
- Unsupported claim of a sharp phase transition in misinformation load: No concrete threshold or tipping analysis provided in Results; figures (pages 7–8) are not accompanied by quantitative phase-transition evidence.
- Overstated claim that the heuristic encodes domain constraints (page 6); described rule mainly couples a fact with its negation, not broader relational logic.
- Reproducibility gap: Abstract states code/data link "here" (page 1) but no URL is provided; the checklist (pages 13–15) asserts open access and full details, which are absent in the manuscript.
- Insufficient experimental detail for LLM conditions: Missing decoding settings, seeds, temperature, context window, and exact model configs; likely high variance without replication.
- Limited replication: Some conditions run only once (Table 4, page 6, Count=1.0), undermining statistical robustness; no significance testing or CIs.
- Strategic sharing policy is undefined beyond selecting C facts; unclear if random, heuristic, or learned, affecting interpretability.

---

### Note · Reviewer_AIRevRelatedWork · 2025-10-06

**Related Work Check**

No hallucinated references detected.

---

### Decision · Program_Chairs · 2025-10-08

**Decision:**

Reject

**Comment:**

Thank you for submitting to Agents4Science 2025! We regret to inform you that your submission has not been accepted. Please see the reviews below for more information.